# Synthetic exploration of sulfinyl radicals using sulfinyl sulfones

Zikun Wang [1,5], Zhansong Zhang[1,5], Wanjun Zhao[1], Paramasivam Sivaguru [1], Giuseppe Zanoni[2],
Yingying Wang[1], Edward A. Anderson [3] & Xihe Bi [1,4✉]

Sulfinyl radicals – one of the fundamental classes of S-centered radicals – have eluded synthetic application in organic chemistry for over 60 years, despite their potential to assemble valuable sulfoxide compounds. Here we report the successful generation and use of sulfinyl radicals in a dual radical addition/radical coupling with unsaturated hydrocarbons, where readily-accessed sulfinyl sulfones serve as the sulfinyl radical precursor. The strategy provides an entry to a variety of previously inaccessible linear and cyclic disulfurized adducts in a single step, and demonstrates tolerance to an extensive range of hydrocarbons and functional groups. Experimental and theoretical mechanistic investigations suggest that these reactions proceed through sequential sulfonyl and sulfinyl radical addition.

[1] Department of Chemistry, Northeast Normal University, Changchun, China. [2] Department of Chemistry, University of Pavia, Pavia, Italy. [3] Chemistry Research Laboratory, University of Oxford, Oxford, UK. [4] State Key Laboratory of Elemento-Organic Chemistry, Nankai University, Tianjin, China. [5]These authors contributed equally: Zikun Wang, Zhansong Zhang. ✉email: bixh507@nenu.edu.cn

Organosulfur compounds are of fundamental importance in synthetic and biological chemistry. They are widespread in molecules such as natural products, proteins, and pharmaceuticals, playing a central role in biology as structural elements and key mediators of biological processes[1–4], and providing vital tools for organic synthesis[5–10]. Many methodologies and sulfur sources have been developed to install a variety of sulfur functionalities into organic molecules[11–14]. Among these, the use of sulfur-centered radicals to access sulfur-containing compounds has attracted considerable attention due to their ease of generation and efficiency of reaction[15–20]. However, in contrast to the popularity of sulfonyl and thiyl radicals, the utilization of sulfinyl radicals in synthetic chemistry has remained unexplored. This is due to the challenges of the inherent properties of sulfinyl radicals: their additions to π-systems are typically reversible, because of the relatively high stability of the sulfinyl radical[21], and they also readily undergo homodimerization to form thiosulfonates (Fig. 1A)[22]. We hypothesized that a dual radical addition/radical coupling strategy might overcome these issues and open up possibilities for the application of sulfinyl radicals in organic synthesis, not least by providing a means to access sulfoxides, which are one of the most important classes of organosulfur compounds[23–25]. Specifically, the coupling of a sulfinyl radical with an in situ generated highly reactive carbon-centered radical could suppress the homo-coupling of the sulfinyl radical and also avoid the generation of a radical at the β-position to the sulfinyl group, thus preventing the undesired β-elimination of the sulfinyl group. Key to the implementation of this strategy is a suitable reagent that under mild conditions can simultaneously release a sulfinyl radical and another radical species of higher reactivity.

Sulfinyl sulfones (Fig. 1B), which are high-valent analogs of disulfides, have been known for over a century[26]. However, their structure and reactivity have only been sporadically investigated[27–30]; the perception that sulfinyl sulfones are unstable, hard-to-handle materials, along with a lack of reliable methods for their synthesis, has deterred research and restricted their occasional use as electrophilic sulfur sources[31–33]. A single report suggested that their thermal decomposition might proceed via homolytic fission, generating two distinct sulfur-centered radicals—a sulfinyl radical and a sulfonyl radical[34]. As sulfonyl radicals are known to undergo facile addition to π-bonds[17–19], sulfinyl sulfones appeared to be well suited for our envisioned strategy.

Here we describe the successful generation and use of sulfinyl sulfones in radical addition/radical coupling reactions with a wide variety of unsaturated hydrocarbons (Fig. 1C). This chemistry offers a strategy for the synthesis of sulfoxide-containing molecules, which are of widespread utility throughout the organic synthesis. Moreover, this reaction provides a simple and efficient method to access high-value disulfurized products, the synthetic utility of which is demonstrated by selective transformation of either the sulfonyl or sulfinyl group into a variety of other building blocks in a controllable fashion.

## Results

### Synthesis, structure, and reactivity of sulfinyl sulfone 1.
We hypothesized that these disulfide derivatives might be prepared by nucleophilic attack of a sulfinate anion on a suitably activated S-electrophile and questioned whether sulfinate salts might serve as the source of both species. After significant efforts, we found that sulfinyl sulfone 1 could indeed be directly prepared by the reaction of sodium *p*-toluenesulfinate with acetyl chloride in chloroform under a nitrogen atmosphere (72% yield, Fig. 1D). The identity of 1 was confirmed by single-crystal X-ray diffraction analysis, which revealed a S–S bond length of 2.230 Å; this is

significantly longer than the S–S bond lengths of other disulfide derivatives[35–38] and may explain the lability of sulfinyl sulfones: while 1 can be stored without decomposition at −18 °C for over 4 months under a nitrogen atmosphere as solid or as the solution in deuterated chloroform, it is sensitive to air and temperatures above 25 °C (Supplementary Fig. 1). We next tested sulfinyl sulfone 1 as a sulfinyl radical source for reaction with phenylacetylene. When 1 was reacted with phenylacetylene at 40 °C within 30 min, a disulfurized adduct 2 was obtained in nearly quantitative yield, the regioselectivity and stereoselectivity of addition being confirmed through single-crystal X-ray diffraction (Fig. 1D).

### Scope of terminal alkynes.
We further questioned whether the synthesis of a sulfinyl sulfone and its subsequent reaction with an unsaturated hydrocarbon could be achieved directly from the sodium sulfinate salt. This indeed turned out to be the case, as the reaction of sodium toluenesulfinate, acetyl chloride, and a variety of alkynes for 30 min under mild heating afforded the corresponding *E*-β-sulfinyl vinylsulfones 2–40 in high yield (Fig. 2A). Terminal (cyclo)alkyl alkynes provided additional products 3–13 in good yield, whereas the reaction with aryl alkynes resulted in near-quantitative yield, regardless of the steric and electronic properties of the substituents (14–29), with the possible exception of 4-nitrophenylacetylene (23, 75%). Equally productive were a naphthyl and two thienyl alkynes (30–32). A conjugated enyne was chemoselectively converted into product 33 with the alkene moiety untouched; however, alkynes featuring functional groups sensitive to acetyl chloride (hydroxyl, amino, carboxyl, amide) required the use of preformed sulfinyl sulfone 1 (12, 13, 27, and 28).

### Scope of internal alkynes.
While the regioselectivity of the reaction with terminal alkynes is dictated by the obvious difference in sterics, for internal alkynes the control on regiochemistry of the products is typically challenging. After ascertaining that our method can deliver a tetrasubstituted disulfurized alkene in good yield, as demonstrated with the *E*-2-butenyl product 34, the reaction protocol was tested with aryl alkyl alkyne substrates, delivering alkenes 35–40 in high yield and exceptional regioselectivity for the product featuring the sulfinyl group positioned adjacent to the aromatic ring, irrespective of the nature of the functional groups on the arene or alkyl chain. In particular, bromoalkene 40, further amenable for functionalization, was obtained as a single isomer. In all cases, the exquisite *Z*-stereoselectivity remained confirmed.

### Scope of alkenes.
We next turned our attention to the disulfurization of monosubstituted alkene and obtained products 42–47 with high efficiency and regioselectivity. Even ethylene gas could be converted to sulfinyl sulfone 41 at atmospheric pressure in good isolated yield. Equally successful was the methodology with a range of acyclic and (hetero)cyclic 1,1-disubstituted alkenes, resulting in products 48–63. The structural determination of 49 by single-crystal X-ray diffraction confirmed the regioselectivity of addition, in analogy to what was observed with terminal alkynes and arylalkyl alkynes. As small carbocyclic and heterocyclic rings have been increasingly included in the design of pharmaceutical interest for their favorable physicochemical properties, as opposed to $sp^2$-hybridized scaffolds[39–41], disulfurized products such as 49 and 53–55 could be attractive building blocks in medicinal chemistry, especially in consideration of the potential further derivatization of the two distinct sulfur-based moiety, as discussed later. 1,2-Disubstituted linear and cyclic alkenes were also suited to the transformation,

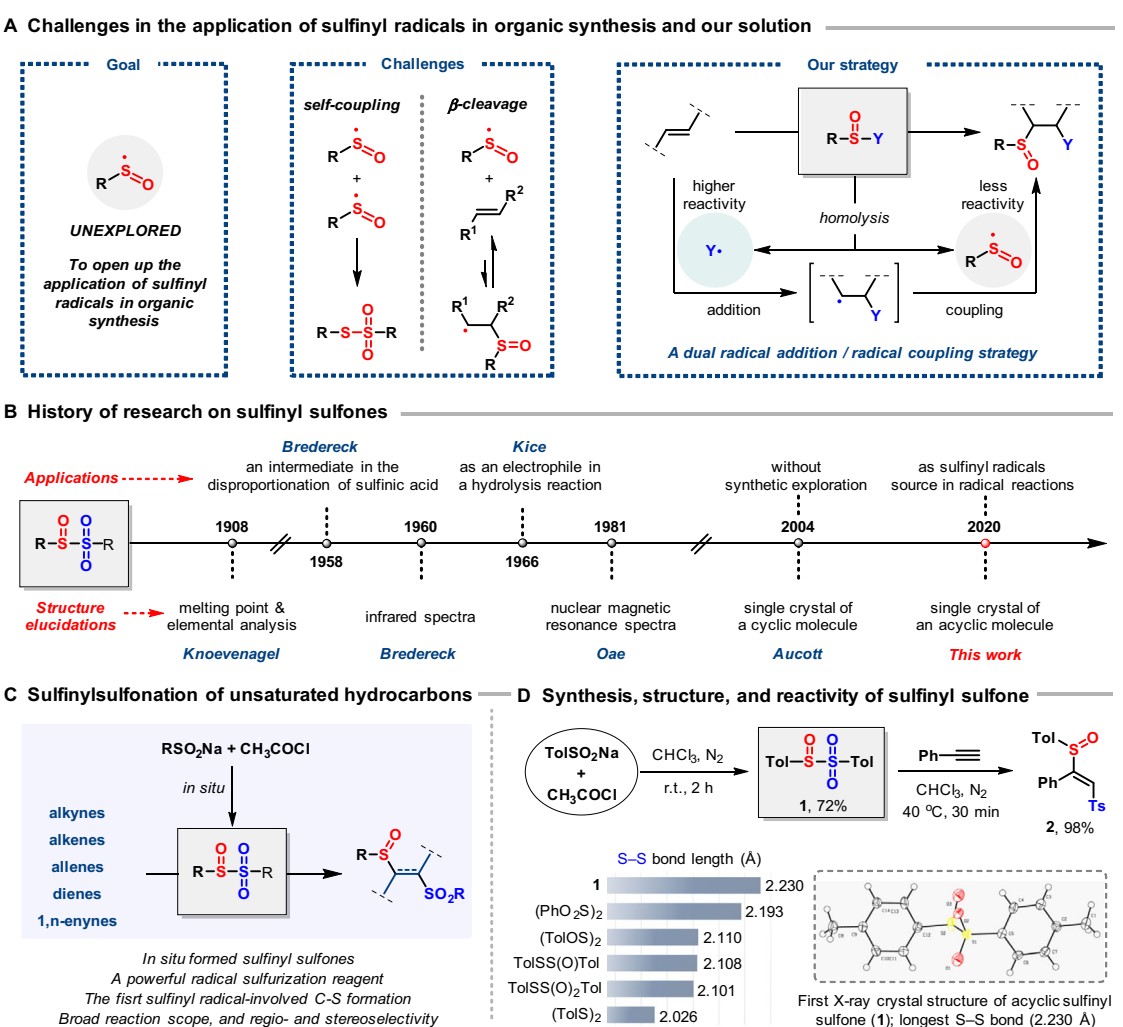

**Fig. 1 The research history of sulfinyl sulfones and their application as a precursor of sulfinyl radical. A** Design of a strategy for the application of sulfinyl radicals in organic synthesis; **B** the discovery history of sulfinyl sulfones, with emphasis on the limited synthetic access and applications; **C** sulfinylsulfonation of various unsaturated hydrocarbons with in situ formed sulfinyl sulfones; **D** The synthesis, single-crystal X-ray structure of sulfinyl sulfone **1**, comparison of the S–S bond length (Å) with that of other analogous compounds, and the sulfinylsulfonation of phenylacetylene.

delivering adducts **64–69** in generally high yield. However, for almost all terminal alkenes and unsymmetric internal alkenes, the sulfinylsulfonation products exhibit poor stereoselectivity. Finally, even 1,3- and 1,2-dienes (allenes) underwent the addition with complete regiocontrol, affording the respective 1,4-addition products **70–72** with up to 6:1 *E/Z* selectivity and allylic sulfones **73–78**, respectively, in moderate-to-good yield.

**Scope of 1,*n*-enynes**. As a consequential development for our sulfinylsulfonylation methodology, we questioned whether it could be adopted in radical cascade cyclizations of 1,*n*-enynes, which notoriously enable the preparation of a variety of carbocyclic and heterocyclic compounds[42]. Such an approach would inevitably present the challenge of controlling the site-selectivity of the addition of either sulfur component onto the enyne moiety. When tested with our protocol, a few probe enyne substrates successfully undertook the targeted radical cascade reaction, delivering products with chemoselectivity and regioselectivity that strictly depended on the substitution pattern of the substrate (Fig. 2B). In particular, the cyclization of 1,6-enynes with a trisubstituted double bond and capped by a phenyl group on the alkyne yielded six-membered endocyclic vinyl sulfones as single regioisomers (**79–81**). The structure of compound **79** was

confirmed by single-crystal X-ray diffraction. Conversely, the regioselectivity of addition was completely reversed for 1,6-enynes with terminal alkynes, giving five-membered exocyclic vinyl sulfones (**82–84**). A complete switch in chemoselectivity was observed for 1,6-enynes with a methylidene moiety, whereby the sulfonyl group was added to the alkene and the sulfinyl substituent to the alkyne. Phenyl alkynes delivered five-membered exocyclic vinyl sulfoxides (**85–87**), while terminal alkynes reversed the regioselectivity of the cyclization, affording six-membered endocyclic vinyl sulfoxides (**88–90**). Overall, this radical cascade enables the efficient formation of one C–C bond and two C–S bonds in one step and offers a powerful means for the construction of sulfur-containing carbocycles and heterocycles with exceptional control over both ring size and substituent regioselectivity.

**Scope of arynes**. Arynes are a class of highly reactive intermediates, generated in situ from certain precursors such as the most commonly used 2-(trimethylsilyl)phenyl triflates, and have rich reactivity, including multi-component reaction, aryne relay reaction, σ-bond insertion, cylcoaddition, and so on[43]. However, the radical reaction of arynes is quite rare, probably due to the low concentration and the high reactivity of both aryne and radical

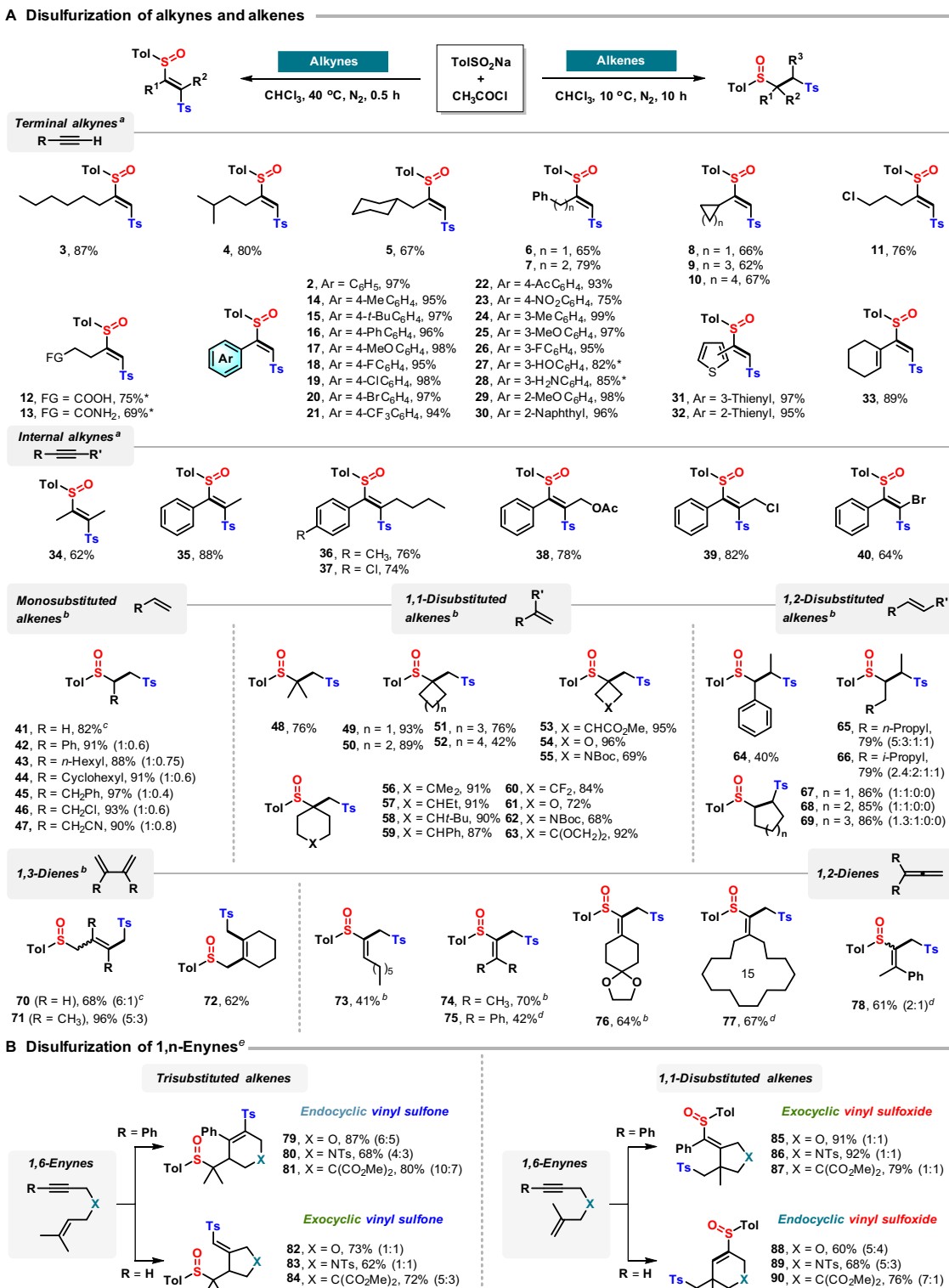

**Fig. 2 The sulfinylsulfonation of unsaturated hydrocarbons. A** Sulfinylsulfonation of alkynes and alkenes; **B** cascade sulfinylsulfonation of enynes. All yields are isolated yields. Diastereomeric ratios are shown in parentheses where appropriate. Amounts of reagents and solvent: alkyne/alkene/enyne (0.3 mmol), TolSO$_2$Na (1.8 mmol), CH$_3$COCl (1.2 mmol), 0.1 M in CHCl$_3$, N$_2$. [a]40 °C, 30 min. [b]10 °C, 10 h. [c]Reaction conditions for gaseous olefins: TolSO$_2$Na (1.8 mmol), CH$_3$COCl (1.2 mmol), CHCl$_3$ (3 mL), 10 °C, 20 h, ethylene/1,3-butadiene (1 atm); the yield of product was calculated using TolSO$_2$Na as the standard. [d]40 °C, 5 h. [e]Room temperature, 8 h. *Reaction conditions for the synthesis of **12**, **13**, **27**, and **28**: alkyne (0.3 mmol), **1** (0.45 mmol), 0.1 M in CHCl$_3$, N$_2$, 40 °C, 0.5 h.

**Fig. 3 The sulfinylsulfonation of arynes.** Reaction conditions: arynes (0.5 mmol), **1** (0.5 mmol), CsF (1.5 mmol), 18-crown-6 (1.5 mmol), DME (3 mL), 25 °C, N₂, 3 h. ᵃ0 °C, N₂, 1 h. ᵇ**99**, **101**, or **103** (0.5 mmol), m-CPBA (0.75 mmol), DCM (3 mL), 25 °C, 1 h.

species generated in situ in the reaction system[44]. We therefore examined the tolerance of arynes as an unsaturated hydrocarbon in our methodology. In order to minimize the interference of reactants, the sulfinyl sulfone **1** was prepared separately and then employed in the reaction with 2-(trimethylsilyl)aryl triflates in the presence of CsF and 18-crown-6. We were pleased to find that both benzyne and naphthalyne precursors smoothly underwent the sulfinylsulfonation reaction to afford the corresponding disulfurized aromatics **91–99**, **101**, and **103** in good-to-excellent yield (Fig. 3). The structure of **91** was confirmed with the help of single-crystal X-ray diffraction. The unsymmetric aryne precursors resulted in a 1:1 ratio of regioisomer mixture, which could be normalized through the oxidation leading to vicinal disulfone compounds **100**, **102**, and **104**.

**Scope of sodium sulfonates and synthetic utility**. Further investigation of the reaction scope with respect to the sulfinate revealed that alkylsulfinates, as well as arylsulfinates with either electron-donating or electron-withdrawing groups, were readily converted into adducts **105–119** in excellent yield (Fig. 4A). Alkyl sulfoxides and alkyl sulfones are widely found in drugs, such as

tinidazole, apremilast, armodafinil, armodafinil, and fulvestrant[1, 3]. In general, these moieties are prepared by oxidation of sulfides; the disadvantage of this method is that it can be difficult to control the oxidation states of organosulfur—a mixture of sulfoxides and sulfones is often obtained, and there is a risk of undesired oxidation of other functional groups[3]. Our method avoids these problems by directly accessing desulfurized compounds containing alkyl sulfinyl and alkyl sulfonyl groups without the need for redox manipulations. Application of the sulfinylsulfonation methodology to the functionalization of alkyne functionality in selected natural products afforded disulfurized products **120–125** in high yield (Fig. 4B). Further, the sulfinylsulfonation of intrinsic alkyne functionality in drugs such as erlotinib and clodinafop-propargyl ester also proved to be suitable and afforded the corresponding adducts **126** and **127** in high yield (Fig. 4C). These results demonstrate the utility of the sulfinylsulfonation method in functionalizing pharmaceutically relevant molecules. Moreover, the sulfinylsulfonation of aryne has been applied to the preparation of vortioxetine (Fig. 4D), indicating that the reaction has potential application in the preparation of medicines.

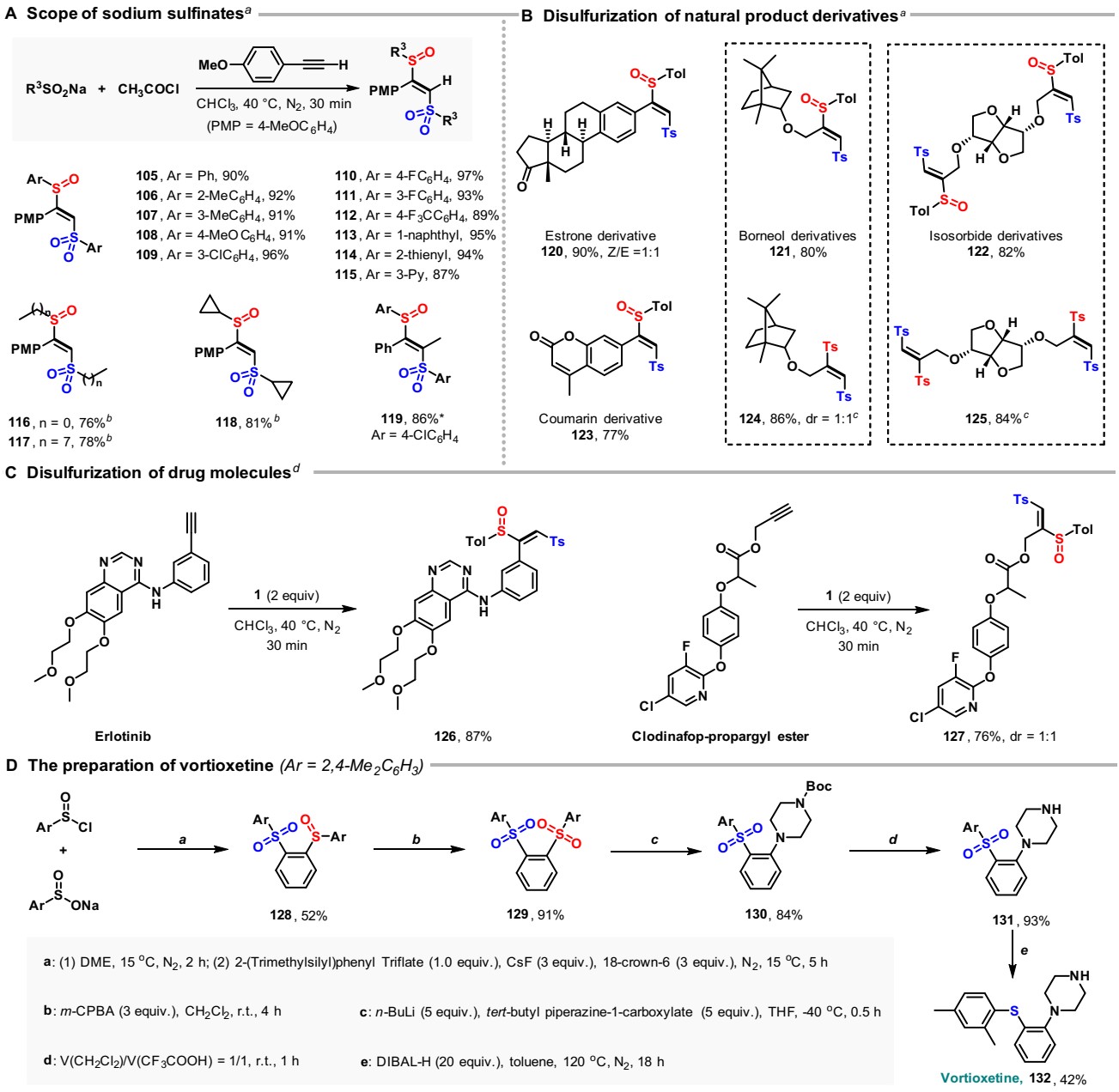

**Fig. 4 The scope of sodium sulfonates and the application of sulfinylsulfonation in late-stage functionalization and synthesis of drug molecules. A** Variation of the sodium sulfinate. *Prop-1-yn-1-ylbenzene was used as the substrate; **B** disulfurization of natural product derivatives; **C** disulfurization of drug molecules. All yields are isolated yields. Reaction conditions for the synthesis of **105**–**123**: alkyne (0.3 mmol), sodium sulfinate (1.8 mmol), acetyl chloride (1.2 mmol), 0.1 M in CHCl₃, N₂. [a]40 °C, 30 min. [b]10 °C, 12 h. [c]**121** or **122** (0.5 mmol), *m*-CPBA (0.75 mmol), DCM (10 mL), room temperature, 1 h. [d]Erlotinib or Clodinafop-propargyl ester (0.5 mmol), **1** (1.0 mmol), CHCl₃ (3.0 mmol), 40 °C, N₂, 0.5 h. **D** The preparation of vortioxetine.

**Large-scale synthesis and further transformations.** Our sulfinylsulfonation protocol can be easily scaled up to more than 100-fold, as demonstrated by the uneventful preparation of a multi-gram quantity of compound **2** in 85% yield after recrystallization (Fig. 5). β-Sulfinyl vinylsulfones are newly synthesized compounds, and the synthetic utility of these now readily accessible disulfurized products was showcased by a large number of transformations. For instance, the selective functionalization of either the sulfonyl or sulfinyl motif in **2** was explored. Treatment of **2** with sodium hydride (NaH) results in an acetylenic sulfone **133** by eliminating the sulfinyl group. 1,8-Diazabicyclo[5.4.0]undec-7-ene as the base affords a product **134** by the selective

reduction of the sulfonyl group. The reaction of **2** with SmI₂ in tetrahydrofuran selectively leads to the desulfinated product **136**, whereas SmI₂ in hexamethylphosphoric triamide induces desulfonation and gives product **135**. Nucleophilic addition/elimination reactions of the sulfinyl group in **2** with amines, azide, and alkoxides give products **140**–**145**, whereas sulfide and selenide ions displace the sulfonyl group to give compounds **137**–**139**. In both cases, inversion of stereochemistry is observed at the double bond (as confirmed by ¹H-¹H nuclear Overhauser effect spectroscopy analysis of **137** and **140**). Use of Grignard reagent (EtMgBr) as the nucleophile results in the nucleophilic substitution of the sulfinyl group to give compounds **136** and **146**, whereas an eliminative

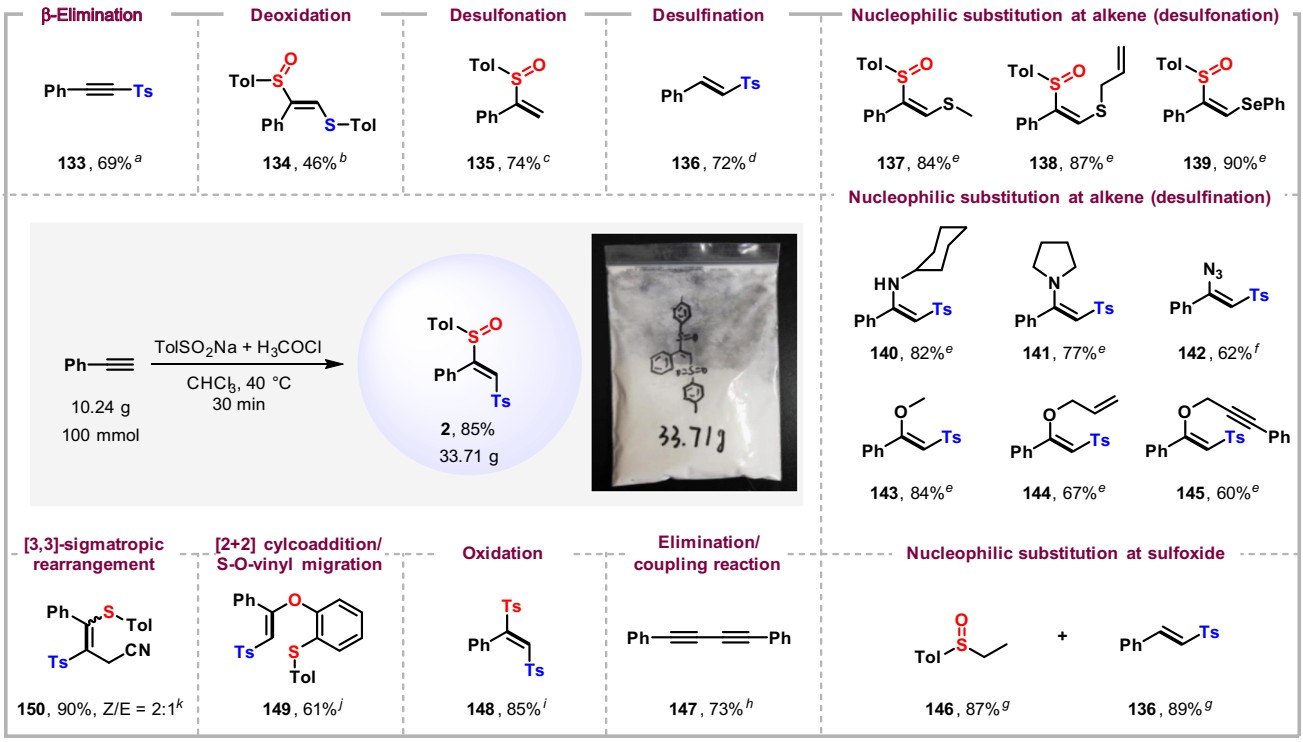

**Fig. 5 Large-scale synthesis and further transformations.** All yields are isolated yields. The amount of **2** is 0.5 mmol. [a]NaH (0.75 mmol), CH$_3$CN (3.0 mL), 25 °C, 6 h. [b]DBU (5.0 mmol), toluene (3 mL), 100 °C, 6 h. [c]Methanol (5.0 mmol), hexamethylphosphoric triamide (HMPA, 2.5 mmol), SmI$_2$ in THF (0.1 M, 25 mL), THF (1 mL), −20 °C, N$_2$, 2 h. [d]SmI$_2$ in THF (0.1 M, 25 mL), methanol/THF (v/v = 1/1, 1 mL), 25 °C, N$_2$; then, 60 °C, 3 h. [e]NaH (0.75 mmol), RXH (0.75 mmol), CH$_3$CN (3 mL), 25 °C, 5 h. [f]NaN$_3$ (5.0 mmol), DMSO (3 mL), 25 °C, 0.5 h. [g]EtMgBr (1.0 mmol), THF (3 mL), 25 °C, N$_2$, 3 h. [h] EtMgBr (1.0 mmol), Ni(acac)$_2$ (0.005 mmol) THF (3 mL), 25 °C, N$_2$, 3 h. [i]m-CPBA (0.75 mmol), DCM (10 mL), 25 °C, 1 h. [j]CsF (3.0 mmol), 2-(trimethylsilyl)phenyl trifluoromethanesulfonate (1.5 mmol), CH$_3$CN (5 mL), 55 °C, 12 h. [k]Tf$_2$O (1.0 mmol), DABSO (1.25 mmol), CH$_3$CN (5 mL), −30 °C, 2 h.

coupling product **147** is produced in the presence of Ni(acac)$_2$. Oxidation of the vinyl sulfoxide with *m*-CPBA gives disulfone **148**. The sulfinyl group reacts with benzyne affording a rearrangement product **149** via [2 + 2] cycloaddition/S-O-vinyl migration cascade[45]. Treatment of **2** with trifluoromethanesulfonic anhydride in acetonitrile leads to a product **150** by [3,3]-sigmatropic rearrangement reaction[46]. Note that some core structures of these products are present in the drug and natural products. For example, the 1,2-disulfonylehene moiety in **148** is the key structure of dimethipin, the 1,3-butadiyne unit in **147** is found as the core structure in natural products such as bupleurotoxin, lobetyolin, enanthotoxin, etc., and the β-sulfonyl enamine in **140–142** is observed in adociaquinones.

**Proposed mechanism.** A proposed mechanism for the sulfinylsulfonation is shown in Fig. 6A, which begins with activation of the sulfinate salt with acetyl chloride to form intermediate (**I**). This intermediate undergoes reaction with a further equivalent of the sulfinate to give the sulfinyl sulfone **1**. No reaction with phenyl acetylene was observed in the absence of acetyl chloride, confirming its essential role as an activator. Homolytic fission of the S–S bond in **1** generates sulfonyl radical **A** and sulfinyl radical **B**. The formation of these two radicals was confirmed by electron paramagnetic resonance experiments with the addition of free-radical spin-trapping agent 5,5-dimethyl-1-pyrroline *N*-oxide (Supplementary Fig. 2). In addition, we also found thiosulfonate as a by-product by the result of sulfinyl radical homocoupling. Based on our experimental observations of regioselectivity, the

more electrophilic radical **A** is proposed to undergo addition to the unsaturated hydrocarbon, leading to the (more stable) carbon-centered radical **II**, which is then captured by the sulfinyl radical **B** to afford product **2**. In support of the proposal of a radical process, the sulfinylsulfonation of phenylacetylene was completely suppressed in the presence of 2,2,6,6-tetramethylpiperidine 1-oxyl, whereas the use of cyclopropylethylene as substrate led to ring-opened product **151** (52%).

**Density functional theory (DFT) rationalization.** The proposed mechanism was studied using DFT calculations at the SMD-B3LYP/6-31+G(d,p) level of theory (Fig. 6B; for additional details, see Supplementary Figs. 3–6). After a mildly endergonic cleavage of the S–S bond in reagent **1**, the addition of the sulfonyl radical **A** to phenylacetylene via **TS1** ($\Delta G^{\ddagger}$ = 7.6 kcal mol$^{-1}$) is favored by $\Delta\Delta G^{\ddagger}$ = 10.7 kcal mol$^{-1}$ over the addition of the sulfinyl radical **B**, which would proceed via **TS1′** ($\Delta G^{\ddagger}$ = 18.3 kcal mol$^{-1}$)—a highly endergonic and reversible process to give radical **II′** ($\Delta G^0$ = +17.1 kcal mol$^{-1}$). The expected higher electrophilicity of **A** is reflected by the higher charge on the sulfur atom in **A** (1.27, calculated by natural population analysis), in contrast to **B** (0.61), thus favoring the attack of **A** over **B** on the alkyne, to give the stabilized radical **II**. The approach of the sulfinyl radical **B** to **II** is sterically directed to give the (*E*)-disulfurized product **2** via **TS2**, $\Delta G^{\ddagger}$ = 18.5 kcal mol$^{-1}$, a lower energy pathway than that leading to (*Z*)-disulfurized product **2-1** via **TS6**, $\Delta G^{\ddagger}$ = 22.4 kcal mol$^{-1}$, in agreement with the experimental findings where no (*Z*)-disulfurized product has ever been detected. Inclusion of dispersion

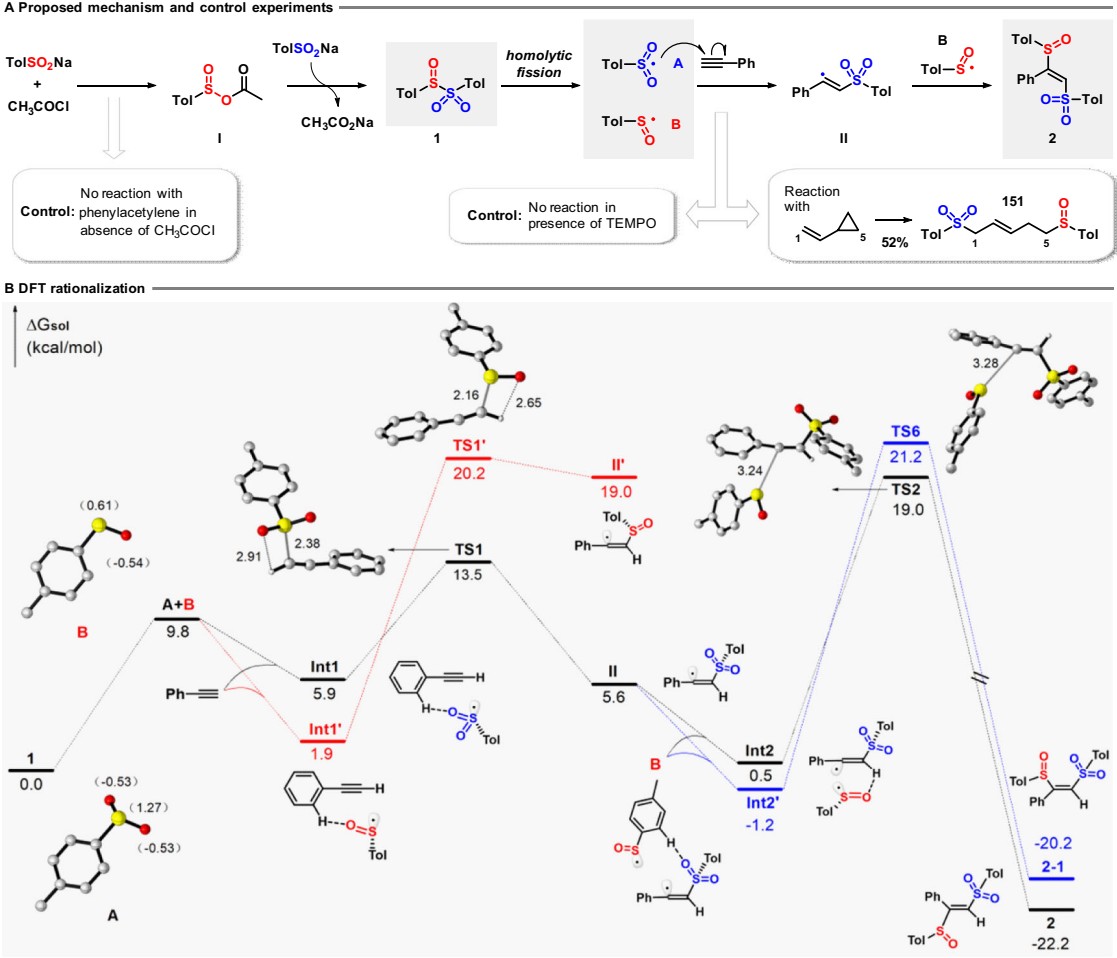

**Fig. 6 Proposed mechanism and theoretical calculation research. A** Proposed mechanism and experimental support for a radical pathway; **B** DFT calculations and graphical representation of the proposed mechanistic pathway. The energies are in kcal mol$^{-1}$ and represent the relative free energies calculated at the SMD-B3LYP/6-31+G(d,p) level in chloroform. The preparation of **1**: TolSO$_2$Na (1.8 mmol), CH$_3$COCl (1.2 mmol), CHCl$_3$ (3 mL), N$_2$, rt, 2 h. The radical suppression reaction: phenylacetylene (0.3 mmol), TolSO$_2$Na (1.8 mmol), CH$_3$COCl (1.2 mmol), TEMPO (0.9 mmol), CHCl$_3$ (3 mL), N$_2$, 40 °C, 30 min. The radical clock reaction: cyclopropylethylene (0.3 mmol), TolSO$_2$Na (1.8 mmol), CH$_3$COCl (1.2 mmol), CHCl$_3$ (3 mL), N$_2$, 10 °C, 10 h.

effects by using the B3LYP-D3 functional leads to similar energies and conclusions (see Supplementary Fig. 7).

## Discussion

In conclusion, we have described a strategy for the utilization of sulfinyl radicals in organic synthesis, overcoming the known tendency to undergo homo-coupling and β-cleavage. The reaction of unsaturated hydrocarbons by a radical mechanism with in situ generated sulfinyl sulfones revealed the potential to harness these compounds as a source of disulfurized organic molecules. The reaction scope spanned from cascade processes to derivatization of scaffolds found in natural products, affording a wide range of organosulfur compounds, which have broad potential use in organic synthesis, pharmaceutical, and materials research. In addition, this dual radical addition/radical coupling concept for the generation and use of sulfinyl radicals can serve as a general model for the development of other sulfur reagents.

## Methods

**The procedure for the synthesis of sulfinyl sulfone 1**. In a Schlenk tube, CH$_3$COCl (1.2 mmol) and sodium *p*-toluene sulfinate (1.8 mmol) were dissolved in CHCl$_3$ (3 mL). The reaction mixture was stirred at room temperature for 2 h. Upon completion, petroleum ether (15 mL) was poured into the reaction mixture. The mixture was filtered and solid were collected. The resulting crude product was

purified by a mixture of petroleum ether and ethyl acetate (petroleum ether/ethyl acetate = 10/1, 3 × 10 mL) to afford sulfinyl sulfone **1** as a white solid in 72% yield. (All the above operations were performed in the glove box.)

**General procedure for the sulfinylsulfonation of alkynes**. In a Schlenk tube, sodium sulfinates (1.8 mmol) and alkyne (0.3 mmol) were dissolved in CHCl$_3$ (3 mL) under a N$_2$ atmosphere, then CH$_3$COCl (1.2 mmol) was added. The reaction mixture was stirred at 40 °C for 30 min. Upon completion, saturated sodium bicarbonate (30 mL) was added and the reaction was extracted with dichloromethane (2 × 30 mL), washed with brine (30 mL), dried over Na$_2$SO$_4$, and concentrated in vacuo. The residue was purified by column chromatography on silica gel with a gradient eluent of petroleum ether and ethyl acetate to provide the desired product.

**General procedure for the sulfinylsulfonation of alkenes**. In a Schlenk tube, sodium sulfinates (1.8 mmol) and alkene (0.3 mmol) were dissolved in CHCl$_3$ (3 mL) under a N$_2$ atmosphere, then CH$_3$COCl (1.2 mmol) was added. The reaction mixture was stirred at 10 °C for 10 h. Upon completion, saturated sodium bicarbonate (30 mL) was added and the reaction was extracted with dichloromethane (2 × 30 mL), washed with brine (30 mL), dried over Na$_2$SO$_4$, and concentrated in vacuo. The residue was purified by column chromatography on silica gel with a gradient eluent of petroleum ether and ethyl acetate to provide the desired product.

**General procedure for the sulfinylsulfonation of arynes**. In the glove box, sulfinyl sulfone **1** (0.5 mmol), CsF (1.5 mmol), and 18-crown-6 (1.5 mmol) were dissolved in dimethyl ether (3 mL) in a Schlenk tube, then 2-(trimethylsilyl)aryl

trifluoromethanesulfonate (0.5 mmol) was added. The reaction mixture was stirred at room temperature for 3 h. Upon completion, saturated ammonium chloride (30 mL) was added and the reaction was extracted with dichloromethane (2 × 30 mL), washed with brine (30 mL), dried over $Na_2SO_4$, and concentrated in vacuo. The residue was purified by column chromatography on silica gel with a gradient eluent of petroleum ether and ethyl acetate to provide the desired product.

## Data availability

The X-ray crystallographic coordinates for structures reported in this study have been deposited at the Cambridge Crystallographic Data Centre (CCDC), under deposition numbers 1994377, 1864547, 2003800, 2003798, 2058384, and 1907354. These data can be obtained free of charge from The Cambridge Crystallographic Data Centre via www.ccdc.cam.ac.uk/data_request/cif. Complete experimental procedures and compound characterization data are available in the Supplementary Information or from the corresponding author upon request.

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

## Acknowledgements

Financial support was provided by NSFC (21871043, 21961130376), Department of Science and Technology of Jilin Province (20180101185JC, 20190701012GH, 20200801065GH), and the Fundamental Research Funds for the Central Universities (2412019ZD001, 2412019FZ006). E.A.A. thanks the EPSRC for support (EP/S013172/1). X.B. and E.A.A. thank Royal Society for a Newton Advanced Fellowship (NAF R1 191210).

## Author contributions

Z.W. and Z.Z. contributed equally to this work. Z.W. and X.B. conceived the strategy, designed the investigation, analyzed the data, and together with E.A.A., G.Z. and P.S. discussed the results and drafted this manuscript. Z.W., Z.Z., W.Z., P.S. and Y.W. performed the experiments and the calculations.

## Competing interests

The authors declare no competing interests.

**Additional information**

