## [Peer Review File · Nature Communications]

REVIEWER COMMENTS

Reviewer #1 (Remarks to the Author):

I would like to start by congratulating the authors on a very nicely and scholarly executed study. Although the strategy for the preparation of sulfinyl radicals from sulfinyl sulfones may not be completely new, the synthetic applications described by the authors are novel and useful: the usually unstable sulfinyl sulfones readily engage various unsaturated hydrocarbons in disulfurization reactions. The reaction exhibits broad scope and generally excellent yields (although it is not clear whether derivatization of existing drug molecules shown in Figure 4 is of value - the alkyne moiety present in them is critical for their activity), and the products can undergo further transformations as exemplified in Figure 5. The radical nature of the proposed mechanism is supported by several control experiments and a DFT computational study. All in all, this is a creative and practical method of functionalizing unsaturated hydrocarbons, and I look forward to seeing its applications in synthesis.

Valery V. Fokin

Reviewer #2 (Remarks to the Author):

In this work, Bi and co-workers reported a combined experimental and computational study on the generation and synthetic application of sulfinyl radical using sulfinyl sulfones. Systematic exploration of the synthetic application of sulfinyl radical, particularly with pi-bonds, were performed. The dual functionalization reactions reached good efficiency and satisfying scope. In addition, DFT calculations were performed to reveal the reaction mechanism of sulfinyl radical addition on alkyne. Overall I think this work is on the borderline of Nature Communications.

Here are a few detailed major issues:

1. The synthetic application of the sulfinyl radical-functionalized product is really poor. The authors should provide a few convincing cases that the developed methodology is useful in the synthesis of bioactive molecule. The radical functionalization of alkene, alkyne and aryne are not surprising.

2. For the DFT calculations, the authors should try a few additional functionals to confirm the reliability of the computed free energy profile, especially considering the open-shell systems. The dispersion correction is now a standard requirement for B3LYP calculations. The authors should also re-compute the free energy profile with B3LYP-D method.

3. It is necessary to compute the closed-shell, open-shell singlet and triplet states of diradical species Int2 and Int2' and the subsequent bonding processes.

4. Charge and spin states of every calculated species should be included in supporting information.

5. The control experiments are not clearly presented in Fig. 6A. Please provide the details of reaction outcome for each tested control experiment.

Reviewer #3 (Remarks to the Author):

This is a very interesting paper, describing the first example of a low temperature formation and use of sulfinyl radicals in synthesis. Part of it is inspired by seminal work by Kice published nearly 50 years ago - and it is indeed surprising that more hasn't been made of sulfinyl radical chemistry in the interim. There is a good breadth of applications and details of further diversification and transformations of the alkyl and alkenyl sulfoxide sulfone products. In effect this is a series of papers pulled into one large disclosure. Is this appropriate for Nature Communications? Certainly I feel the simple method to generate and use sulfinyl radicals is worthy of a high impact communication.

I feel the paper is well written, and the spectroscopic information adequate (although it is a shame IR data is not included) - The inclusion of X-ray structures of a variety of compounds also helps.

I feel on balance I recommend this work for publication in Nature Communications.

A POINT-BY-POINT RESPONSE TO REVIEWER COMMENTS

Manuscript ID: NCOMMS-21-16476-T

Title: Synthetic exploration of sulfinyl radicals using sulfinyl sulfones

Author(s): Zikun Wang, Zhansong Zhang, Wanjun Zhao, Paramasivam Sivaguru, Giuseppe Zanoni, Yingying Wang, Edward A. Anderson, Xihe Bi

Dear Reviewers,

Thank you very much for your suggestion. We have revised this manuscript according to your comments. The corrections in detail were given in the revised manuscript and were highlighted in yellow color. The detailed revision was listed as follows:

Reviewer 1:

I would like to start by congratulating the authors on a very nicely and scholarly executed study. Although the strategy for the preparation of sulfinyl radicals from sulfinyl sulfones may not be completely new, the synthetic applications described by the authors are novel and useful: the usually unstable sulfinyl sulfones readily engage various unsaturated hydrocarbons in disulfurization reactions. The reaction exhibits broad scope and generally excellent yields (although it is not clear whether derivatization of existing drug molecules shown in Figure 4 is of value - the alkyne moiety present in them is critical for their activity), and the products can undergo further transformations as exemplified in Figure 5. The radical nature of the proposed mechanism is supported by several control experiments and a DFT computational study. All in all, this is a creative and practical method of functionalizing unsaturated hydrocarbons, and I look forward to seeing its applications in synthesis.

Valery V. Fokin

Response: We thank Prof. Fokin for taking the time and effort to review our manuscript and gave positive comments on our work.

Reviewer 2:

In this work, Bi and co-workers reported a combined experimental and computational study on the generation and synthetic application of sulfinyl radical using sulfinyl sulfones. Systematic exploration of the synthetic application of sulfinyl radical, particularly with pi-bonds, were performed. The dual functionalization reactions reached good efficiency and satisfying scope. In addition, DFT calculations were performed to reveal the reaction mechanism of sulfinyl radical addition on alkyne. Overall I think this work is on the

borderline of Nature Communications.

Here are a few detailed major issues:

1. The synthetic application of the sulfinyl radical-functionalized product is really poor. The authors should provide a few convincing cases that the developed methodology is useful in the synthesis of bioactive molecule. The radical functionalization of alkene, alkyne and aryne are not surprising.

Response: Thank you very much for the reviewer's suggestion. To further demonstrate the potential application of our sulfinylsulfonation protocol in drug synthesis, we have applied the sulfinylsulfonation of aryne in the synthesis of vortioxetine. The results were added into Fig. 4D.

2. For the DFT calculations, the authors should try a few additional functionals to confirm the reliability of the computed free energy profile, especially considering the open-shell systems. The dispersion correction is now a standard requirement for B3LYP calculations. The authors should also re-compute the free energy profile with B3LYP-D method.

Response: According to the reviewer's suggestions, the additional dispersion correction functionals M06-2X and B3LYP-D3 were used to calculate the radical addition reaction of carbon-centered radical **II** and sulfinyl radical **B**.

We also re-compute the free energy profile with the B3LYP-D3 method.

The calculated results suggested the inclusion of dispersion corrections with the M06-2X and B3LYP-D3 functionals do not change the results. Furthermore, using the B3LYP-D3 method, the rate-determining step is changed to homolytic fission of the sulfonyl sulfone **1** ($\Delta\Delta G^\ddagger = 19.7$ kcal/mol). However, since sulfonyl radical **A** and sulfonyl radical **B** are easily generated at low temperatures (0 K), B3LYP-D3 appears to overestimate the dissociation energy. Therefore, the results calculated by the B3LYP method were listed in the main text of

manuscript, while the results calculated by the B3LYP-D3 method were listed in the Supporting Information.

3. It is necessary to compute the closed-shell, open-shell singlet and triplet states of diradical species **Int2** and **Int2'** and the subsequent bonding processes.

Response: Geometry optimizations of **Int2** and **Int2'** were performed without any symmetry constrained by using RB3LYP, ROB3LYP, and UB3LYP methods. The results are as follows:

	Int2	Int2'
RB3LYP	 bonding product Energy: -1872.3601316 a.u.	 bonding product Energy: -1872.057018 a.u.
ROB3LYP	 bonding product Energy: -1872.3601311 a.u.	 bonding product Energy: -1872.057114 a.u.
UB3LYP	 Energy: -1872.076879 a.u.	 Energy: -1872.078171 a.u.

In the closed-shell and open-shell singlet optimizations, the adduct **Int2** and **Int2'** are easy

to form bonding products, so the diradical species cannot be obtained under these calculations. Besides, the energy of **Int2'** calculated by UB3LYP is the lowest. Therefore, the spin multiplicity of diradical species **Int2** and **Int2'** is 3 (triplet state).

There are no analytic 2nd derivatives for the restricted open-shell method in the Gaussian software to optimize the transition state. Here, we use RB3LYP and UB3LYP methods to optimize the geometries of **TS2** (trans addition) and **TS6** (cis addition). The results are as follows:

	TS2	TS6
RB3LYP	 Energy: -1872.047425	 Energy: -1872.158507
UB3LYP	 Energy: -1872.033533	 Energy: -1872.032239

The results show that the energy of the singlet state is lower than that of the triplet state in the diradical bonding processes.

The calculation results suggest that the spin multiplicity of diradical species **Int2** and **Int2'** is 3 (triplet state) and the transition states of the subsequent bonding processes are singlet states.

4. Charge and spin states of every calculated species should be included in supporting information.

Response: Thanks for your careful examination. We have added the charge and spin states

of every calculated species in Supporting Information.

5. The control experiments are not clearly presented in Fig. 6A. Please provide the details of reaction outcome for each tested control experiment.

Response: The details of the reaction conditions for each control experiment were added in the caption of Fig. 6.

Reviewer 3:

This is a very interesting paper, describing the first example of a low temperature formation and use of sulfinyl radicals in synthesis. Part of it is inspired by seminal work by Kice published nearly 50 years ago - and it is indeed surprising that more hasn't been made of sulfinyl radical chemistry in the interim. There is a good breadth of applications and details of further diversification and transformations of the alkyl and alkenyl sulfoxide sulfone products. In effect this is a series of papers pulled into one large disclosure. Is this appropriate for Nature Communications? Certainly I feel the simple method to generate and use sulfinyl radicals is worthy of a high impact communication.

I feel the paper is well written, and the spectroscopic information adequate (although it is a shame IR data is not included) - The inclusion of X-ray structures of a variety of compounds also helps.

I feel on balance I recommend this work for publication in Nature Communications.

Response: Thank you very much to the reviewer for taking the time and effort to review our manuscript and gave positive comments on our work.

Some other changes were also made and marked in YELLOW color. Finally, we would like to show our great respects to all the reviewers. Your efforts have improved the quality of this manuscript. We hope that the revised manuscript will reach the level for publication in Nature Communications.

Thank you once again. We are looking forward to hearing from you.

Yours sincerely

Xihe

REVIEWERS' COMMENTS

Reviewer #2 (Remarks to the Author):

The authors have addressed my previous concerns. I support is publication in Nature Communications.